



# A Framework for Improving Data Quality of Thermo-Hygrometer Sensors aboard Unmanned Aerial Systems for Planetary Boundary Layer Research

Antonio R. Segales[1,2,3], Phillip B. Chilson[4], and Jorge L. Salazar-Cerreño[1,3]

[1]The University of Oklahoma School of Electrical and Computer Engineering, Norman, Oklahoma
[2]Cooperative Institute for Severe and High-Impact Weather Research and Operations, The University of Oklahoma, Norman, Oklahoma
[3]Advanced Radar Research Center, The University of Oklahoma, Norman, Oklahoma
[4]Center for Autonomous Sensing and Sampling, The University of Oklahoma, Norman, Oklahoma

**Correspondence:** A. Segales (tony.segales@ou.edu)

**Abstract.** Small Unmanned Aerial Systems (UAS) are becoming a good candidate technology for solving the observational gap in the planetary boundary layer (PBL). Additionally, the rapid miniaturization of thermodynamic sensors over the past years allowed for more seamless integration with small UASs and more simple system characterization procedures. However, given that the UAS alters its immediate surrounding air to stay aloft by nature, such integration can introduce several sources of bias and uncertainties to the measurements if not properly accounted for. If weather forecast models were to use UAS measurements, then these errors could significantly impact numerical predictions and, hence, influence the weather forecasters' situational awareness and their ability to issue warnings. Therefore, some considerations for sensor placement are presented in this study as well as flight patterns and strategies to minimize the effects of UAS on the weather sensors. Moreover, advanced modeling techniques and signal processing algorithms should be investigated to compensate for slow sensor dynamics. For this study, dynamic models were developed to characterize and assess the transient response of commonly used temperature and humidity sensors. Consequently, an inverse dynamic model processing (IDMP) algorithm that enhances signal restoration is presented and demonstrated on simulated data. A few real case studies are discussed that show a clear distinction between the rapid evolution of the PBL and sensor time response. The conclusions of this study provide information regarding the effectiveness of the overall process of mitigating undesired biases and distortions in the data sampled with a UAS and increase the data quality and reliability.

## 1 Introduction

Technological development with respect to instrumentation systems for weather sampling increasingly demands the means to provide greater reliability of the data collected. Furthermore, Lorenz (1972) showed that the results from numerical weather





20 models tend to diverge with long periods of time and differ from reality even with small errors in the initial conditions estimated from measurements. Researchers have been looking for ways to increase the reliability and accuracy of weather measurements, like Mahesh et al. (1997) who successfully implemented a simple method to correct thermal lags from measurements taken with a radiosonde in strong inversions. Radiosondes have the advantage that their sensors are exposed to the medium they are sampling without much disturbances, as opposed to their Unmanned Aerial Systems (UAS) counterparts which produce an

25 inherent turbulent micro-environment around its body (Greene et al., 2018).

 Recent technological advancements have enabled the use of UAS as tools to perform controlled and targeted atmospheric measurements. UAS have paved the way for the development of new strategies for sampling the atmosphere in the past few years. The National Academies of Sciences and Medicine (2018), the National Research Council (2009) and other studies (Hardesty and Hoff, 2012; Geerts et al., 2017) have stressed the importance of the contributions that UAS have made in modern

30 meteorological studies. It is well known that the planetary boundary layer (PBL) is quite under-sampled and that observational gaps limit the ability to accurately estimate the state of the atmosphere; hence, UAS are seen as new opportunities to fill the gap (Bell et al., 2020). In other words, UAS are able to sample regions of the atmosphere that were either not feasible or not possible with other conventional meteorological instruments.

 Despite presenting attractive and unique features, the UAS must still undergo several studies and evaluations before its data

35 can be fully integrated and assimilated into the weather forecast models. Several recent collaborative field experiments, like those described in Barbieri et al. (2019); Koch et al. (2018); Kral et al. (2018) and Jacob et al. (2018) just to cite some of them, have encouraged researchers and engineers to start characterizing and assessing UAS for measuring weather parameters, and identify the challenges for improving weather measurements using UAS. This initiative led to the development of many innovative UAS designs for weather sampling such as shown in Segales et al. (2020); Wildmann et al. (2014a); Reuder et al.

40 (2009), and even envisioning future concepts of operations (Chilson et al., 2019) and research communities (de Boer et al., 2020).

 With the advent of UAS technology for weather sampling, new advanced sampling strategies and signal processing capabilities are becoming possible. The mitigation of slow sensor dynamics and the removal of sensor noise using low cost weather sensors are challenging, but the impacts can be reduced by using the right tools. The inverse dynamic model processing (IDMP)

45 techniques have traditionally been used in control theory for the design of controllers to influence the system's behavior. This modern technique makes use of known physical properties of the sensor to restore the original signal given a sensor reading. To ensure a reliable and proper functioning of the weather sensors, it is important to mitigate sources of error around the UAS by applying strategies discussed in this study, in particular for temperature and humidity sensors. Along with this systematic bias in the measurement comes the challenge of mitigating errors caused by the transient response of the sensor. Slow transient re-

50 sponse in sensors are commonly associated with amplitude attenuation and phase delay of the output signal (measured weather signal) with respect to the input signal (actual weather signal). While the impact of sensor dynamics can largely be neglected when considering static scenarios, measurements should not be considered instantaneous in space and time when the sensor moves through strong gradients (Houston and Keeler, 2018). Several studies have proposed ways to reduce the impact of the

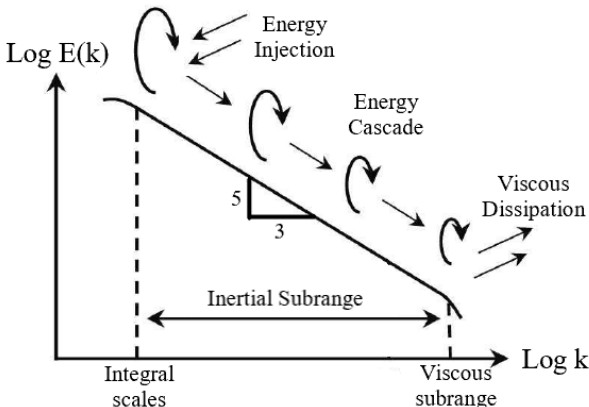

**Figure 1.** Kolmogorov's energy cascade illustration.

sensor transient response for temperature (Dantzig, 1985; Fatoorehchi et al., 2019) and humidity (Wildmann et al., 2014b)
sensors.

The acquisition of weather data using UAS is a newly established challenge in modern meteorology research. The unique
design flexibility and capabilities of the UAS together with the IDMP and signal conditioning tools could greatly reduce
uncertainties in the data. The goal of this project is to improve the quality of the weather data by following a framework
designed around the IDMP method. This will result in a more accurate weather parameter estimates that could, in a near future,
improve data assimilation into weather forecast models and, hence, issue accurate weather warnings. It is critical to provide
forecasters with reliable data in a timely manner to support them in their mission of protecting lives and properties, and help
increasing the nations prosperity.

Considering the above context and problem definition, the following study develops a framework for the characterization
and measurement correction of temperature and humidity sensors with data collected using rotary-wing UAS (rwUAS). This
includes the mitigation of undesired contamination, sensor characterization, and signal conditioning and restoration to improve
the reliability of the weather UAS deliverables.

## 2    Preliminary concepts of the Planetary Boundary Layer

The atmosphere is in a perpetual state of horizontal and vertical motions while constantly evolving day and night. This constant
motion in the atmosphere produces natural weather signals that usually have a high degree of complexity in time and space
(Petty, 2008; Davidson, 2015). However, for the purpose of the demonstrations in this study, the selection of PBL conditions
were narrowed to two particular cases: a well-mixed convective boundary layer (CBL) in windy conditions, and a PBL with
strong temperature and humidity gradients; such as frontal and thermal inversions (FTI) caused by displacement of air masses.
These two PBL states are attractive atmospheric conditions to consider when evaluating weather sensors on UAS because of
several theoretical assumptions that can be made when running simulations.





The CBL condition is ideal to study the small scale turbulence and the high frequency response of the sensors by means of power spectrum and structure function analysis. The energy cascade theory formulated by Kolmogorov is a well proven theory (Kolmogorov, 1941) that can be used to estimate the turbulence energy distribution over a range of spatial scales under locally isotropic conditions. In one of the largest experiments done by Saddoughi and Veeravalli (1994), it was shown that in a turbulent atmosphere with a high Reynolds number the energy cascade decreases with a -5/3 slope in a logarithmic scale and

then tails off downwards in the viscous dissipation region (Figure 1). The approximate relation between the turbulence energy $\Phi_T$ of temperature and the spatial wavenumber $k$ of the temperature signal is given by Tatarskiy (1988):

$$\Phi_T(k_1) \approx 0.25 C_T^2 k^{-5/3}, \tag{1}$$

where $C_T$ is the structure function parameter for temperature. The humidity also has a similar expression as Equation (1) but with a different proportionality constant.

Moreover, the Reynolds number $Re$ of the PBL is typically in the order of $10^6$, and the ratio between large and small spatial scales is given by $l/\eta = Re^{3/4}$. The large separation of scales allows for the inertial subrange (ISR) of turbulent fluctuations to extend for hundreds of meters in length, which would be quite difficult to cover with a UAS within a reasonable time. A workaround to this is to consider horizontally homogeneous CBL conditions. Consequently, the turbulence can be assumed to be "frozen" as it travels across a stationary rwUAS at mean wind speed, or at airspeed relative to a moving fwUAS. This

assumption is the so called Taylor's hypothesis of the frozen-field, and it is of great use in calculating structure functions by converting temporal measurements into spatial measurements.

The definition of the structure function can be found in a vast literature (Gibbs et al., 2016; Kaimal et al., 1976; Kohsiek, 1982). The physical interpretation of structure function is the distribution of turbulent energy over different spatial lags and it is mathematically defined as a two point spatial correlation as follows:

$$D_T^2(r) = \overline{(T(x) - T(x+r))^2} = C_T^2 r^{\frac{2}{3}}, \tag{2}$$

where the overbar represents ensemble averaging, $x$ is the position vector in meters, and $r$ is the separation distance between two samples in space, also called distance lag. If the distance lag $r$ is within the ISR, then the structure function is reduced to the rightmost expression of Equation (2). In the ISR region, the structure function follows a 2/3 slope line in a logarithmic scale.

For the case of FTIs, the PBL undergoes a quick evolution with strong gradients over a short time period where large scale changes of temperature and humidity are dominant over small scales. This is a good scenario for studying the low frequency response of the sensors when flown across the air mass boundaries. Therefore, for this project, the main focus was to create artificial weather signals with strong gradients similar to that of FTIs without much importance on small amplitude and high frequency features. Temperature and humidity changes across a frontal or thermal inversion can be approximated to a ramp

function with rounded corners.





## 3 Weather Sensors Principle and Modelling

The performance optimization of the sensor starts with the available manufacturing technologies according to Farahani et al. (2014). Nowadays, the fabrication of sensors are driven by low-cost circuits, new sensing materials, advances in miniaturization techniques, and modern simulations. Despite that great part of the performance of the sensor can be optimized at a

hardware level, they still come with limitations. Post-processing algorithms can partially overcome these limitations in performance, modelling techniques and digital signal processing being the most popular. Dantzig (1985) showed encouraging first results using a simple first order differential equation to restore signals from a thermocouple, and even describing calibration procedures. Fatoorehchi et al. (2019) used non-linear differential equations based on the Steinhart and Hart (1968) equation for negative temperature coefficients (NTC) thermistors. Their equations comprised of a lumped formulation for temperature

which also included other factors like thermal radiation and power dissipation. Despite its high accuracy, the high complexity of the model and solution makes it unpractical for real-time implementation. Finally, Wildmann et al. (2014b) shows an example of the modelling of a capacitive humidity sensor using the diffusion equation and effectively applying an inverse model to correct the measurements. Ideas from this study were borrowed to develop the IDMP proposed in this report and it also served as a guidance to develop an IDMP variant for the bead thermistor.

### 3.1 Basics of temperature and humidity sensors

Commonly used temperature and humidity sensors for UAS are mainly variants of the bead thermistor type and capacitive type sensors, respectively (Barbieri et al., 2019; Kral et al., 2018). These type of sensors are considered payload friendly for small UAS given their compact size and lightweight characteristics. Both of these temperature and humidity sensors work under very similar principles. Basically, the heat flux (diffusion) inside the sensor's material will lead to a thermal (water vapor

concentration) equilibrium with the surrounding medium after a finite time. In fact, the differential equation that describes great part of their behavior has the exact same form for both sensors which is given by Pletcher et al. (2013):

$$\frac{\partial U}{\partial t} = k \left( \frac{\partial^2 U}{\partial x^2} + \frac{\partial^2 U}{\partial y^2} + \frac{\partial^2 U}{\partial z^2} \right). \tag{3}$$

This is called the heat equation for the case of temperature and diffusion equation for the case of water vapor concentration. Farahani et al. (2014) explains that numerous parameters have an influence on the response time of a sensor, such as the

geometry of the sensing element, the inherent thermal/water diffusivity of the sensing element, the thickness of the protective layers, and even the ambient temperature and humidity itself. Equation (3) encompasses most of these characteristics and factors and; hence, can be effectively used as a model to compensate for errors.

In particular, the selected sensors for this study are the iMet-XF bead thermistor from InterMet Systems and the HYT-271 capacitive humidity sensor from Innovative Sensor Technology (IST). This is for the sake of providing an example case

with known sensor characteristics and specifications, the overall modelling and method description does not lose its generality. Additionally, it is assumed the sensors are sufficiently aspirated (airflow of $> 5$ m s$^{-1}$) as per recommendation of the manufacturers so that self-heating effects are diminished.

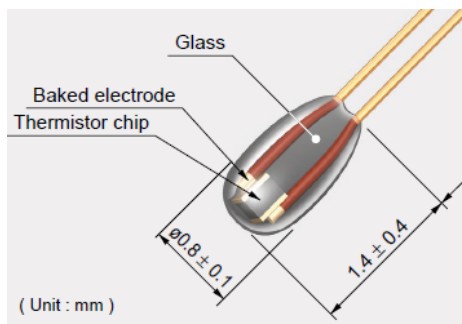

**Figure 2.** Closeup of the iMet-XF bead thermistor with dimensions and composition. Image provided by International Met (InterMet) Systems.

## 3.2 Sensor dynamic modelling

In control theory, the method of finite difference is a commonly used numerical solution for differential equations and it is the main foundation of the IDMP method. Finite difference equations are powerful tools that can be used to create mathematical models to describe the behaviour of physical systems. The method is an approximation to the derivative which is represented by the derivative taken over a finite interval around a given point. Additionally, assumptions must be made to reduce the complexity of the model and work within a linear regime. The dynamics of the sensor can be further studied after deriving the mathematical model. It can be used to trace back and restore the original signal that produced the sensor measurements as long as the inverse of the model exists and is stable.

### 3.2.1 Forward model of temperature sensor

The iMet-XF bead thermistor shape and dimensions are shown in Figure 2, the probe tip can be approximated to the shape of a sphere with radius $R = 0.4$ mm. Given that heat fluxes can propagate anywhere around the sensor, the problem becomes three-dimensional in space. The spherical symmetry helps to significantly reduce the degree of complexity of the model to a one-dimensional case along the radius (Momoh et al., 2013). It was assumed that the heat propagates radially from the surface of the spherical glass in contact with the air all the way down to the core where the sensing element is located. Therefore, the problem can be seen as a heat transfer problem with spatial temperature gradients inside the bead thermistor. Accordingly, the differential equation for the bead thermistor is given by Equation (3) in spherical coordinates:

$$\frac{\partial T(r,t)}{\partial t} = \alpha \left( \frac{\partial^2 T}{\partial r^2} + \frac{2}{r} \frac{\partial T}{\partial r} \right); 0 \leq r \leq R, \, t \geq 0, \tag{4}$$

where the internal temperature $T$ is a function of time $t$ and space $r$ which is the radial distance from the center to a given point within the sphere, and $\alpha$ is the thermal diffusivity of the material. The boundary conditions are $T(R,t) = T_{air}$ and $T(0,t)$ is mirrored. The sphere was then divided into N layers with thickness $\Delta r = R/N$. Finite difference method was applied to the spatial derivatives of Equation (4) and the singularity at $r = 0$ was solved by using L'Hopital rule. The following system of





finite difference equations in space was obtained,

$$\frac{\partial T}{\partial t} = \beta \left( -2T_N + \frac{N-1}{N} T_{N-1} \right) + \beta \frac{N+1}{N} T_{air} \qquad\qquad r = R,$$

$$\frac{\partial T}{\partial t} = \beta \left( \frac{i+1}{i} T_{i+1} - 2T_i + \frac{i-1}{i} T_{i-1} \right) \qquad\qquad 0 < i\Delta r < R,\ i = 2,...,N-1$$

$$\frac{\partial T}{\partial t} = 3\beta \left( T_2 - T_1 \right) \qquad\qquad r = 0,$$

where $\beta = \alpha/\Delta r^2$. A more detailed derivation of these equations can be found in Momoh et al. (2013). As a result, the system of equations is a linear time invariant (LTI) system that can be transformed to state-space representation of the form:

$$\frac{\partial x}{\partial t} = Ax + Bu$$

$$y = Cx + Du,$$

where $x$ is the state variables each one representing the temperature at each layer, $u$ is the surface temperature input signal, and $y$ is the sensor's output signal (sensor reading or measurement at the core). The matrices A and B can be easily obtained by inspection from the finite difference equations.

The goal is to determine the actual temperature of the medium based on temperature readings at the core. By computing the inverse model it is possible to trace back the temperature at each layer from the core to the surface of the sensor in a stable manner. In order to do so, the C vector must be a weighted average skewed towards the core. The D matrix was chosen in a way so that the DC gain ($K$) of the system is equal to one; $K = D - CA^{-1}B = 1$.

### 3.2.2 Forward model of humidity sensor

In the case of the HYT-271 capacitive humidity sensor, the dynamics are mainly produced by the diffusion of water vapor concentration from the surface in contact with the air into the polymer. Figure 3 shows the sensing element configuration and the boundary conditions around it. In contrast to the bead thermistor, the capacitive sensor can be treated as a one-dimensional problem since the water vapor only exists just above the sensing element surface and it propagates along the normal to the surface.

Horizontal concentration gradients were considered to be negligible given that the thickness of the polymer is small enough to prevent horizontal propagation (Wildmann et al., 2014b). As a result, the differential equation for the capacitive humidity sensor is given by Equation (3) in 1-D Cartesian coordinates:

$$\frac{\partial c(x,t)}{\partial t} = D \frac{\partial^2 c}{\partial x^2}; \ 0 \le x \le L, \ t \ge 0, \tag{5}$$

where $c$ is the water vapor concentration in parts per million volume (ppmv), $D$ is the diffusivity coefficient of the water vapor in the polymer, and $L$ is the thickness of the polymer. The boundary conditions were defined as $c(L,t) = c_{air}$ while $c(0,t)$ is mirrored. The polymer film was then divided into N layers with thickness $\Delta x = L/N$. Following a similar reasoning and steps



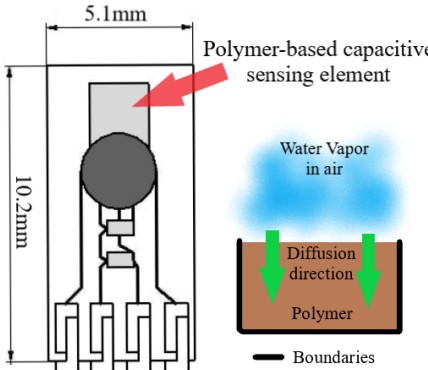

**Figure 3.** Closeup of the IST HYT-271 capacitive sensor with dimensions (left) and sensing element configuration (right). Left image taken from datasheet.

as applied with the bead thermistor model, the system of finite difference equations is as follows:

$$\frac{\partial c}{\partial t} = (-2c_N + c_{N-1}) + \lambda c_{air} \qquad\qquad x = L,$$

$$\frac{\partial c}{\partial t} = \lambda (c_{i+1} - 2c_i + c_{i-1}) \qquad\qquad 0 < i\Delta x < L,\ i = 2,...,N-1$$

$$\frac{\partial c}{\partial t} = \lambda (c_2 - c_1) \qquad\qquad x = 0,$$

where $\lambda = D/\Delta x$. A more detailed derivation of the above equations can be found in (Wildmann et al., 2014b). Again, the resulting system of equations is an LTI system that can be transformed to state-space representation. The matrices A and B can be determined by inspection from the finite difference equations. The output $y$ of the system is equal to the average of all the water vapor concentrations in each layer. This is because the capacitance is measured across the entire polymer film and not

just one particular spot as compared to the bead thermistor. Therefore, the parameter C is a length-N row vector with elements equal to $1/N$ and it maps the state variables to the output resulting in the sensor measurement.

The HYT-271 humidity sensor comes hard-coded to output relative humidity values. The model only works with water vapor concentration in ppmv units. Therefore, the relative humidity input must be converted to water vapor concentration and then back to relative humidity after applying the model. The following equations for the conversion were taken from McRae (1980) who states that the error involved in using these equations over a temperature range of -50° C and 50° C is less than 0.5%:


$$c = 10^4 H \frac{P_s}{P_a},$$

$$P_s = P_a \exp(13.3185t - 1.9760t^2 - 0.6445t^3 - 0.1299t^4),$$

$$t = 1 - \frac{373.15}{T_a},$$

where $H$ is relative humidity in percentage, $P_s$ is saturation vapor pressure in millibars, $P_a = 1013.25mb$ is the standard
atmospheric pressure, $T_a$ is the ambient temperature in Kelvin.



## 4 Experimental Design for Sensor Characterization

The geometry and boundary conditions were defined in the forward sensor models presented in Section 3.2. The sampling period of the sensor must also be known at this point, both iMet-XF and HYT-271 has sampling periods equal to $\Delta t = 0.1$ sec. The remaining parameters to be defined in the models are the thermal (or water) diffusivity $\alpha$ (or $D$), the width of the sensing material $R$ (or $L$) and the thickness of the layers $\Delta r$ (or $\Delta x$). Unfortunately, these parameters are usually not in the datasheet and even kept as a trade secret by the manufacturer. However, these parameters are associated with the time response of the model and they can be adjusted to approximate the time response of the real sensor (Wildmann et al., 2014b). Therefore, the remaining parameters can be obtained empirically in the lab through experimentation. In addition, please note that the response time is defined as the time required for the sensor output to change from its initial state to a final fixed value within a tolerance, typically defined as 98% of the final value.

In an effort to establish guidance for the sensor characterization on UAS, Jacob et al. (2018) conducted experiments where weather UASs were flown across a pseudo-step change in temperature and humidity from an air conditioned room to the outdoor environment. Although their goal was to measure the sensor time response with the effects of the UAS on the sensors altogether, a different approach was taken for the presented framework where the problem was divided into two parts. First, the sensors were isolated and characterized independently of the UAS body through experimentation. Second, sensor siting on the UAS and sampling techniques were investigated to minimize the external disturbances on the measurements.

Ideally, the time response is measured by stimulating the sensor with an ideal step function. However, step functions of temperature and humidity are not possible in real-world conditions. Instead, Li et al. (2018) used a ramp function to model the thermodynamic shock and found that the error of assuming an ideal step function is less than 10% if the transition time from the initial to the final state is less than the time response of the sensor. According to the iMet-XF bead thermistor datasheet, the sensor response time can be less than 1sec. Therefore, it is a requirement to have a mechanism to quickly move the sensors from one side of the shock to the other.

There are several external factors that influence the physical aspects of the sensor that must be taken into account for the experiments. In particular, the capacitive humidity sensor is mainly affected by the ambient temperature as explained by Farahani et al. (2014). This is because the porosity and thickness of the sensor's polymer significantly changes with temperature. This means that there is no universal water vapor diffusivity $D$ that can be used for a wide range of ambient temperature. Therefore, the shock experiments for the humidity sensor must be performed under multiple air temperature conditions to create a lookup table of values. After a UAS flight, the mean temperature of the profile can be used to determine the diffusivity value from the lookup table by interpolation. Conversely, the thermal diffusivity $\alpha$ of the bead thermistor does not significantly change with humidity as measured by Tsilingiris (2008). Consequently, only one shock experiment would be enough to compute a universal thermal diffusivity $\alpha$ for the bead thermistor.

After conducting the experiments for sensor characterization, the collected data is used to adjust the sensor models parameters by using a model parameter optimization technique called Differential Evolution (Das and Suganthan, 2011). Simulated step functions with amplitudes equal to the real thermodynamic shocks from the experiments are fed to the digital models with




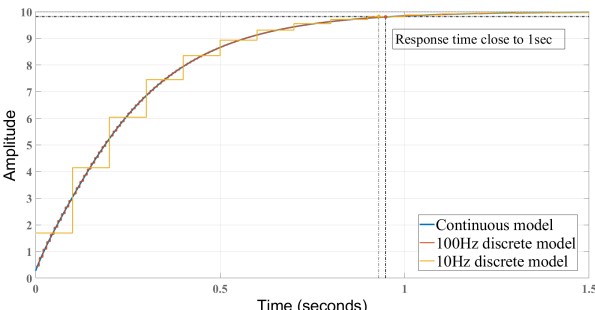

**Figure 4.** Step response of the continuous and discrete models of the bead thermistor.

equal sampling rate as the real sensors. The model parameters can be initialized through a reasonable guess, and the subsequent parameters are computed by iterative point-wise comparisons with the real data. Basically, this becomes a curve fitting problem by minimizing the square root of the mean of the squared errors (RMSE) cost function across the data set:

$$RMSE = \sqrt{\frac{\sum (U_{mod} - U_{obs})^2}{N}} < \epsilon, \tag{6}$$

where $U_{mod}$ is the predicted value of the model, $U_{obs}$ is the real observation, $N$ is the number of data points and $\epsilon$ is the desired tolerance. The thermal (water vapor) diffusivity are slightly tweaked on every subsequent iteration until meeting the tolerance criteria, and thus obtaining the final diffusivity coefficient for the given conditions.

Finally, the sensor time response can be assessed quantitatively by examining the final RMSE value. If the RMSE value remains high after several iterations, it means that the sensor transient response is highly non-linear compared to the linear model and; hence, a higher uncertainty in the measurement correction process is expected. Figure 4 shows an example of how the step response of the continuous and discrete models of a bead thermistor should look like with a response time close to 1sec. Notice how important is to have a high sampling rate logger in order to accurately capture the dynamics of the sensor.

## 5   Considerations for Sensor Placement on UAS

Acquiring precise measurements of the air temperature and humidity is particularly challenging due to disturbances arising from multiple sources around the UAS. Insufficient radiation shielding, exposure to mixed turbulent air from the propellers, and electronic self-heating are the main factors in contributing to temperature and humidity data contamination according to Greene et al. (2018, 2019) and Islam et al. (2019). A quick logical solution is to make an extension arm from the main body of the UAS and place the sensor package outside of the turbulent air created by the UAS. However, this approach comes with major problems such as increased resistance to rotation (or inertia) and exposure to strong aerodynamic forces that could produce flight instability. Therefore, the integrated design must be balanced in such a way that neither of the systems gets compromised.



The rwUASs are the most vulnerable platforms since they usually remain stationary in the air or travel at very low speeds (mainly vertical profiles), allowing for the turbulent air volume to wrap around the rwUAS. Greene et al. (2018) searched for the best-possible location for the thermodynamic sensors within the turbulent air volume. They measured variations in temperature right below a rwUAS propeller and found that disturbances were minimum at 1/3 from the tip of the propeller. Despite the
encouraging results, the study did not take into account the heat advection coming from the UAS body. Islam et al. (2019) took a step further by mounting pipes with air inlets situated far apart from the main body while still using the rwUAS rotors to produce mechanical aspiration. They reported good agreement between ascent and descent profiles in no wind conditions, but also showed limitations with the flight stability and heat advecting into the pipes downwind of the UAS. Segales et al. (2020) took an innovative approach by modifying the autopilot code to compute wind direction estimates and command the rwUAS
to turn into the wind. This way, the wind itself compresses the turbulent envelope in front of the rwUAS making it shallow enough to place the sensors closer to the rwUAS's body. This was demonstrated using flow simulations by Segales et al. (2020) and also through observations in the field by Greene et al. (2019) and Bell et al. (2020).

An important consideration that is common to both rwUAS and fwUAS is the material selection for the radiation shield around the weather sensors. Greene et al. (2019) has shown that the sun radiation greatly affects the temperature and humid-
ity measurements. They noticed an unusual pattern in the readings during flights in a day with scattered clouds despite that the rwUAS had a shield around the sensors. Basically, the sun radiation heats up the surface of the shield and, consequently, increases the temperature of the surrounding air by heat conduction. The heated air is then aspirated across the sensors. Materials with low thermal conductivity can help reducing heat conduction to the air and; hence, mitigate the contamination. For instance, aerogel can be a great candidate material that could almost completely isolate the sensors from the sun radiation.
Also, radiant-barrier paint can be used to coat the current shield designs and prevent surfaces from loading with heat.

## 6 Implementation of the IDMP

At this point, two models were introduced, one describing the bead thermistor sensor dynamics and another describing the capacitive humidity sensor dynamics. An experimental design for sensor characterization and model validation were also presented, and some of the limitations were reviewed. In an effort to reinforce the concepts just introduced, and to demonstrate
the actual implementation of the IDMP method, this section will fully describe the procedures of the IDMP for correcting sensor measurements.

### 6.1 General procedures and limitations

In order to detail and formulate the IDMP restoration technique, it will be beneficial to start by summarizing a basic general procedure:

    1. Identify the sensor noise floor from the actual measurements by inspecting its power spectral density (PSD) and suppress the noise using a zero-phase lowpass filter.



2. Apply the inverse sensor model to the filtered measurements after adjusting the model parameters based on the weather conditions and sensor characterization.

3. Re-apply the filter from step 1 to the restored signal to filter out any amplified noise.

4. Quantify the improvement by inspecting the time series and spectral response of the signal before and after applying the inverse sensor model.

5. Validate the correction by comparing the structure function with the theoretical 2/3 slope for locally isotropic turbulence in the ISR.

Although the general procedure seems to be robust in the sense that it includes a validation step, the validation step only works for CBL weather conditions as explained in Section 2. Furthermore, step 5 is feasible only if the atmosphere is assumed to be horizontally homogeneous and locally isotropic without any rapid atmospheric evolution. Additionally, the flight patterns are limited to steady hover in windy conditions and horizontal transects assuming Taylor's hypothesis of frozen field holds true. For the case of flights across FTIs, only steps 1 to 3 are feasible and the results can be trusted based on prior testings and calibrations.

## 6.2 System description and tuning

The signal conditioning of the raw sensor measurements is necessary to extract a more accurate representation of the atmospheric parameters. The challenge is to remove the unwanted distortion and contamination caused by slow sensor dynamics and sensor noise, respectively. If the considerations from Section 5 are correclty implemented, then the effects of the UAS and its immediate surroundings on the sensor measurements are considered to be small and can be ignored. The type of lowpass filter chosen for noise removal was a non-causal zero-phase finite impulse response (FIR) digital filter. This kind of filter has the advantage that it does not introduce phase delay to the output signal with respect to the input signal, which is only possible in an offline processing.

An adaptive contamination removal algorithm was not developed for the IDMP method. Instead, the cut-off frequency of the lowpass filter was manually tuned on a case by case basis. The PSD of the raw sensor measurement was used to assess the dynamics of the sensor, identify the noise floor and define the cut-off frequency of the lowpass filter. Moreover, the logic used to identify distortion and contamination in the measurements taken in CBL conditions is as follows:

– If the slope of the PSD is greater than -5/3 at high wavenumbers, then the region is considered to be contaminated by sensor noise. Therefore, it must be removed using the lowpass filter before going through the restoration phase.

– Otherwise, the measurement is considered to be attenuated and distorted by the the slow sensor dynamics. Consequently, the bad trend is corrected in the restoration phase and the new PSD approximates the desired -5/3 slope line.

The next step is to run the conditioned signal through the core of the IDMP method, the inverse sensor model itself. The transfer function $H(z)$ of the sensor model was computed by taking the z-transform of the state-space system of the sensor model. Figure 5 shows an example of the frequency response of the continuous and discrete transfer function $H$ of a capacitive





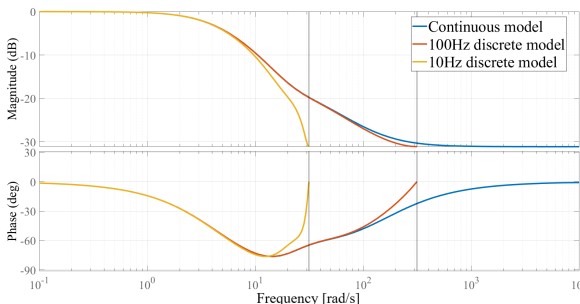

**Figure 5.** Bode diagram of the sensor model. Magnitude and phase are shown on top and bottom, respectively.

humidity sensor with response time of about 4sec. The continuous system is the approximation to the analog behavior of the

sensor and can be used to simulate the "real" sensing element. The discrete system is a "sampled" version of the continuous system with sampling rate equal to that of the real sensor, or the analog-to-digital converter (ADC) to be precise. The truncation effect that comes with sampling a system is what produces the mismatch in the frequency response at high frequencies which is seen in Figure 5. The only way to improve the frequency response of the discrete system is by increasing the sampling rate of the ADC to better capture the dynamics of the sensing element.

The inverse system of $H(z)$ exists and it is stable if and only if $H(z)$ is minimum phase, meaning that all the poles and zeros are within the unit circle of the z-plane. Subsequently, the transfer function of the inverse sensor model is obtained by simply taking the inverse of $H(z) = G^{-1}(z)$. Even though the process of finding the inverse sounds simple and straightforward, special attention must be given to the poles of $G(z)$. The resulting poles of $G(z)$ might end up being too close to the unit circle of the z-plane which could cause instability and oscillations with high frequency input signals. The stability parameter for the

presented models is defined by Pletcher et al. (2013) as $\phi = \frac{\gamma \Delta t}{\Delta x^2} < 0.5$ where $\gamma$ is an intrinsic parameter of the sensor, such as the thermal and diffusivity coefficients. The equation shows how fast the sampling rate must be in order to precisely capture the dynamics of the sensor within a small spatial interval. Given that the sampling rate of the sensors are fixed to $\Delta t = 0.1$ sec, than the only way to adjust the poles is by varying $\Delta x$. The model and; hence, its inverse $G(z)$ are stable with large values of $\Delta x$ at the expense of reducing the order of the model and, consequently, the overall accuracy and resolution of the method.

Moreover, notice that the degree of correction made on the sensor measurements is limited by the sensor's sampling rate. If the same sensors are enabled to sample at higher rates, the IDMP will be more effective.

After tuning the parameters and creating a stable system, $G(z)$ was then fed with the filtered sensor measurements to produce the corrected sensor measurement. Finally, the same lowpass filter from the beginning was applied to the corrected signal to remove any amplified noise that survived the process.





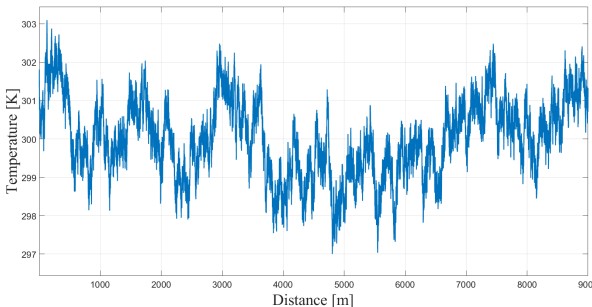

**Figure 6.** Spatial temperature signal in CBL conditions after converting the time-series data given a constant wind speed of $10 \text{ m s}^{-1}$.

## 7 Performance evaluation of the IDMP

The first step to investigate the potential of applying the IDMP for temperature and humidity measurements is to develop a time-series weather signal generator that could be used as a benchmark to evaluate the performance of the proposed framework. From this point, signals made by the generator will be referred as "actual" weather signals, whereas the output signals from the sensor models will be referred as the "measured" signal, and the restored signals from the IDMP will be called "corrected" signals.

### 7.1 Weather signal generation and sensor simulation

As described in Section 2, CBL weather signals from a horizontal transect tend to have a particular PSD with a -5/3 slope in the logarithmic scale. Moreover, assuming horizontal CBL weather signals are produced by a wide-sense stationary (WSS) random process with Gaussian probability density function, then it is possible to generate the artificial weather signals by modifying the PSD of a Gaussian white noise signal to look like Kolmogorov's energy cascade power spectrum. This was achieved by taking the Fourier transform of a white noise signal and dividing the magnitude by its frequency to the power of 5/6 while keeping the phase unchanged. Figure 6 shows an example of an artificially generated CBL weather signal.

For the case of FTI conditions, the focus is mainly on the evaluation of the sensor response in strong gradients with less importance on features with small amplitude and high frequency. Therefore, the FTI weather signals were constructed in a piece-wise manner using straight lines. The horizontal transect of an FTI was modeled using a ramp function, whereas the vertical transect follows a dry adiabatic lapse rate model with a temperature inversion in the middle. Both weather signals were lowpass filtered to smooth out the corners so that it looks more realistic. The gradient (or inversion) strengths, length and altitude scales of the weather signals were adjusted using the simulation results shown by Houston and Keeler (2018) as a reference. Figure 7 shows examples of generated FTI weather signals.

To simulate the sensor measurement process, the sensor model was divided into three parts: the analog sensing element, the ADC discrete sampling and sensor noise generation. The sensing element was simulated using the forward models shown in Section 3.2 with a much smaller sampling period $\Delta t = 0.01$ sec. This is because analog signals can not be generated in a





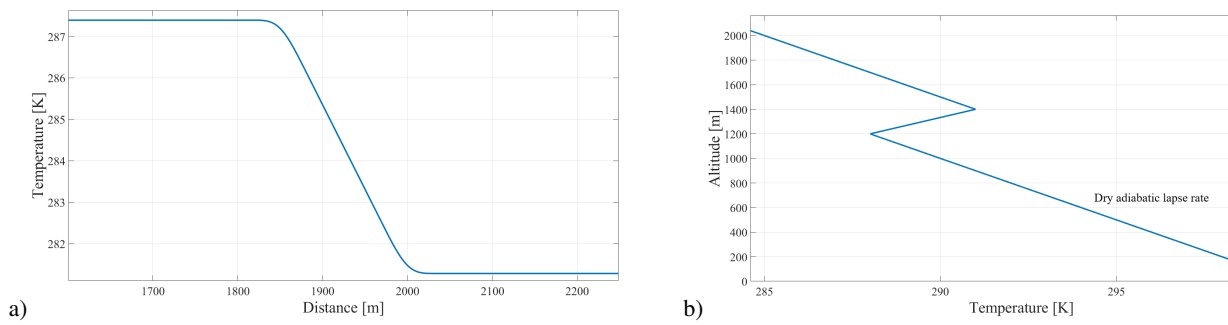

**Figure 7.** a) Temperature change across a frontal inversion (cold front). b) Vertical dry adiabatic lapse rate temperature profile with a strong thermal inversion.

computer and, therefore, the best approximation is to increase the resolution of the discrete model. The actual weather signal was run through the high resolution sensor model to add the effects of sensor dynamics. This signal then goes through the ADC

which down samples the signal to the actual sampling period of the sensor $\Delta t = 0.1$ sec. The down-sampling process may add aliasing which makes it more realistic. Finally, the down-sampled signal gets its characteristic noise floor by adding additive white Gaussian noise (AWGN) to it. The noise amplitude from each sensor was taken from previous steady-state calibrations done in a controlled environment by the Center for Autonomous Sensing and Sampling (CASS) of the University of Oklahoma.

### 7.2   Validation of the IDMP method in simulated CBL conditions

The following measurement validation method for CBL conditions exploits the ISR of turbulent fluctuations theory, described in Section 2, by using the PSD and structure function calculations. In a real-world CBL conditions, the data are collected by conducting horizontal transects or stationary flights in windy conditions at a constant altitude with a rwUAS. In order to keep this document short and because the correction results of temperature and humidity are very similar, results will be shown in an alternated fashion between temperature and humidity.

Following the procedures of the IDMP from start to end, the PSD of the signals are computed and compared in Figure 8. Notice that the cut-off frequency of the lowpass filter was selected near the constant and flat power level of the measured signal, the sensor noise was effectively removed as a result. The effect of the slow sensor dynamics is noticeable as a downward trend with respect to the -5/3 slope line. The IDMP successfully restores the power levels of the measured signal at high frequencies. Next, Figure 9 shows a comparison of the time-series signals, the RMSE of the measured and corrected signals were computed

with respect to the actual weather signal using Equation (6). The time-series plot clearly shows an improvement in the time response of the sensor which is confirmed by the lower RMSE value. Finally, Figure 10 shows results from the two point spatial correlation calculation, namely, the structure function. Assuming locally isotropic turbulence conditions, the deviations of the computed structure function from the theoretical 2/3 slope in the ISR region are indications of the effects of sensor dynamics and sensor noise on the measurement. Moreover, the results can be used to slightly tune the IDMP until getting a best-possible





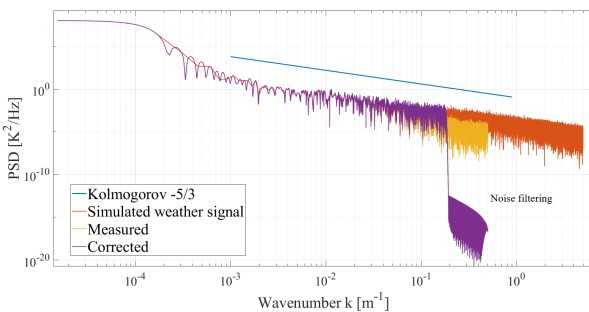

**Figure 8.** Power spectral density of the simulated weather signal in CBL conditions and processed signals using the sensor models. Results from before and after applying the IDMP are shown.

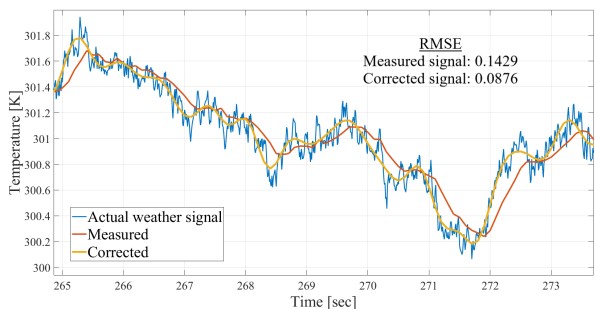

**Figure 9.** Time-series of the simulated weather signal in CBL conditions and processed signals using the sensor models. Results from before and after applying the IDMP are shown.

agreement with the theory. Additionally, all the presented results show that the IDMP method is effectively restoring the signal without any signs of instability and oscillations.

### 7.3 Simulation of flights across strong gradients in FTI conditions

Similar to the CBL simulation, flights across FTIs also exhibits the expected lag in the measurement as a consequence of the sensor time response. Additionally, the lag within this conditions becomes more apparent and sensitive to the relative wind
speed with respect to the UAS. Houston and Keeler (2018) explains that the errors are less when flying at low speeds; however, the observation might not be representative because the weather phenomena might have evolved faster than the observation period. Therefore, it is of significant importance to study the performance of the IDMP method with different speeds across the thermodynamic boundary. For the case of frontal inversions, results from the simulated humidity sensor will be shown since it has a large time response and the correction is more noticeable compared to the faster temperature sensor. Whereas for the
thermal inversion, results from the simulation of the temperature sensor will be shown.

Frontal inversions, such as cold fronts, are usually sampled using fixed-wing UAS (fwUAS). The typical ground speed of a fwUAS in flight is around 25 m s$^{-1}$, while the wind can reach speeds of 20 m s$^{-1}$. Assuming that the fwUAS is able to fly

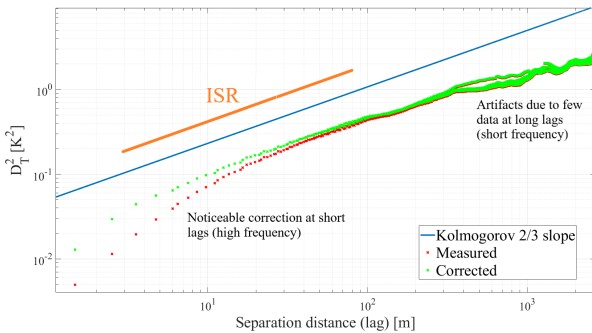

**Figure 10.** Structure function of the simulated weather signal in CBL conditions and processed signals using the sensor models. Results from before and after applying the IDMP are illustrated..

in very windy conditions, then the relative wind across the fwUAS is 45m s$^{-1}$ when flying headwinds. Figure 11a shows the comparison between the actual, measured and corrected humidity in a frontal inversion. Notice how far the measured signal

settles with respect to the air mass boundary, this agrees with the results seen in Houston and Keeler (2018). The corrected signal shows a much better transient response, but some oscillations are present when the weather signal is constant. This is because of some remaining sensor noise leaking into the IDMP where it gets slightly amplified. Additionally, the IDMP can not fully recover the shape of the actual weather signal because of missing parts in the frequency content due to noise filtering, low sampling rate, and poor capturing of the weather dynamics.

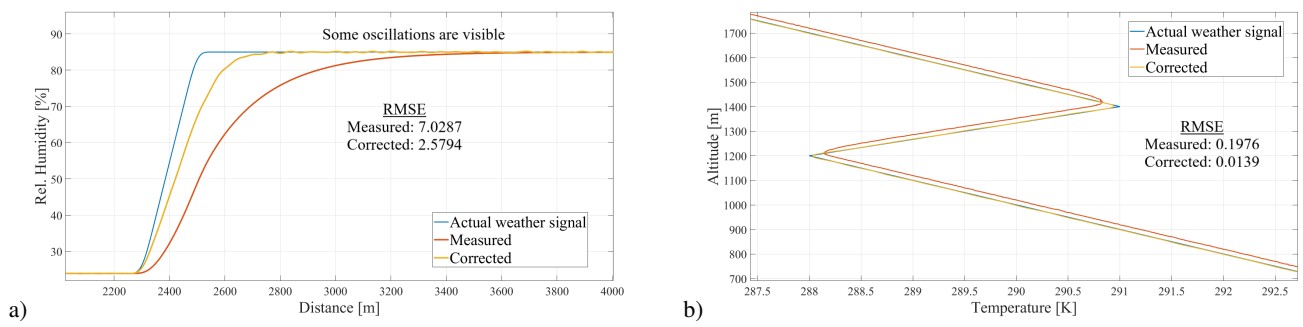

**Figure 11.** Comparison of weather signals in simulated a) frontal inversion conditions with relative wind speed of 45m s$^{-1}$, and b) thermal inversion conditions with climbing rate set to 5m s$^{-1}$. Results from before and after applying the IDMP are illustrated.

Vertical profiles are typically carried out using rwUAS and it is common to encounter thermal inversions aloft. The vertical speed of the rwUAS was assumed to be equal to the radiosonde's climbing speed of about 5m s$^{-1}$, much slower compared with the horizontal speed of a fwUAS. Figure 11b illustrates a comparison between signals in the simulated thermal inversions environment. It can be seen that the IDMP corrects for the constant offset produced by the sensor dynamics not being able to





keep up with the temperature change rate. Once the rwUAS encounters the inversion, the result is almost identical to the frontal

inversion case.

### 7.4 Evaluation with real data

The presented simulation results show the feasibility of the framework and the IDMP technique on measurements taken with a UAS. However, several assumptions were made to produce the models and simulations which may not hold true for real observations in the field. Therefore, to begin exploring the mitigation of slow sensor dynamics and sensor noise for realistic

UAS flights, the IDMP was applied on real data collected using the CopterSonde rwUAS (Segales et al., 2020) from the University of Oklahoma (OU). Two flights were picked from the extensive database made in the past years throughout several field campaigns. These flights were conducted in CBL and FTI weather conditions, respectively, at the Kessler Atmospheric and Ecological Field Station (KAEFS) in Purcell, Oklahoma, USA, located 30 km southwest of the OU Norman campus. It has to be mentioned that the sensors were not properly characterized as described in the presented framework, instead the sensor

models were tuned by trial and error and best guessing the physical parameters of the sensing elements.

In the CBL conditions, the CopterSonde was flown stationary at a constant altitude of 10 m for about 15 min with a mean wind speed of 10.2 m s$^{-1}$. Figure 12a shows a close up of a portion of the measured and corrected relative humidity time-series, while Figure 12b illustrates the degree of correction made by the IDMP. This is noticeable by observing the amount of deviation in the structure function.

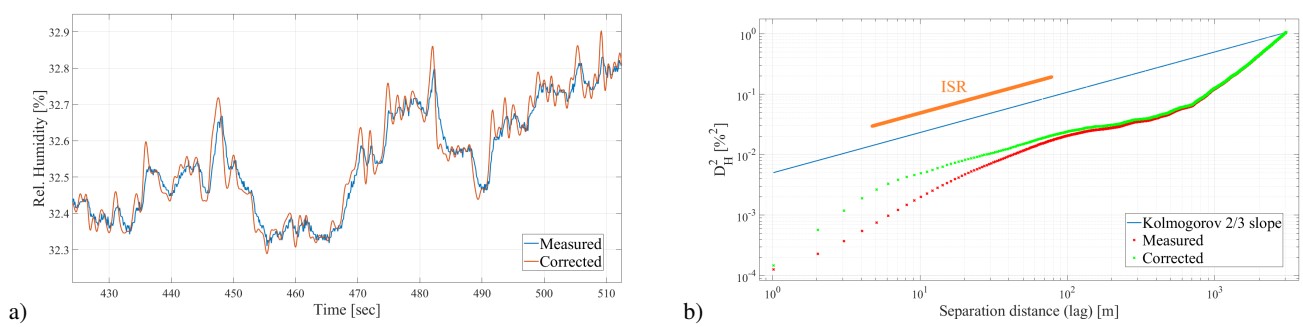

**Figure 12.** a) Comparison of the measured against corrected relative humidity signal. b) Structure function comparison.

In the FTI conditions, the CopterSonde was flown shortly after a cold front moved through KAEFS, leaving a shallow cold pool behind. The climbing rate was set to 3.5 m s$^{-1}$, whereas the descent rate was set to 5 m s$^{-1}$. The CopterSonde was sent to 1300 m above ground level, collecting temperature and humidity data in the ascent and descent legs. The flight took about 10min from take-off to landing. Figure 13a shows a comparison between the measured and corrected vertical profiles of relative humidity. The correction is not very noticeable due to the large spatial scale and the slow vertical speed of the

UAS. However, 13b is a zoomed-in plot of the lower altitude region where the small correction is visible. Assuming that no other environmental factors influenced the measurements, this may confirm that the separation shown by the arrow is in fact an atmospheric evolution and not a result of sensor lag.





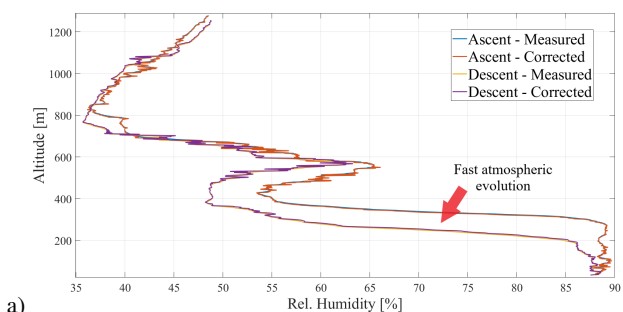

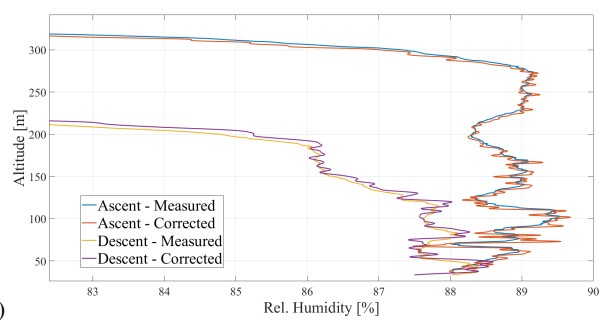

**Figure 13.** a) Relative humidity vertical profile, measured against corrected values. b) Close up of the vertical profile in the lower altitude region.

## 8 Conclusions

This document presented an overview of the general framework and procedures for sensor response and uncertainty mitigation on temperature and humidity measurements collected using a UAS. Important considerations about the effects of the UAS on the sensor measurements were made, in which it is encourage to find solutions that benefit both flight stability and weather sampling accuracy. As a way to complement the mitigation of bias, weather sensor models were developed to investigate and assess the sensor transient response and characterize them through experimentation. This allowed for the implementation of signal restoration techniques such as the IDMP shown in this study. After a brief review of the PBL theory, it has been found ways to corroborate the corrections, like the use of the PSD and structure functions calculations, which gave some validation to the results. Overall, this study presented a framework for the characterization and measurement correction of temperature and humidity sensors with data collected using rotary-wing UAS (rwUAS). The mitigation of undesired contamination, sensor characterization, and signal conditioning and restoration was found to be crucial to improve the reliability of the weather UAS deliverables. The improved weather parameters accuracy can lead to better data assimilation into weather forecast models. Consequently, forecasters can be assured of the improved data quality which helps them to fulfill their mission.

*Data availability.* Data are available upon request to the corresponding author.

*Author contributions.* Conceptualization, A.S. and P.C.; Methodology A.S., P.C., and J.S.; Software, A.S.; Formal analysis, A.S., P.C., and J.S.; Investigation, A.S.; Resources, A.S. and P.C.; Writing - original draft, A.S, P.C., and J.S.; Writing - Review  Editing, A.S., P.C. and J.S.; Supervision, P.C. and J.S.; Funding acquisition, P.C.

*Competing interests.* The authors declare that they have no conflicts of interest.





*Acknowledgements.* This research has been supported in part by the National Science Foundation under Grant No. 1539070 and internal funding from the University of Oklahoma.



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
