# Peer review of "Considerations for Improving Data Quality of Thermo-Hygrometer Sensors aboard Unmanned Aerial Systems for Planetary Boundary Layer Research"

_Atmospheric Measurement Techniques, 2021_

## Referee Comment (RC2)

Review of

**A Framework for Improving Data Quality of Thermo-Hygrometer Sensors aboard Unmanned Aerial Systems for Planetary Boundary Layer Research**

Antonio R. Segales, Phillip B. Chilson, and Jorge L. Salazar-Cerreño

**Recommendation:**
Accept with minor revisions

**Summary:**
The authors present a method for correcting temperature/RH observations collected by small UAS with particular focus on addressing errors associated with sensor response. The method is described thoroughly and includes the motivations and justifications for the decisions made. The method is tested using both synthetic and actual data and performance is evaluated using appropriate techniques.

Overall, this is a solid manuscript describing a method for addressing a common source of measurement error applicable to many observation platforms. I have one "major" comment listed below with minor comments listed according to a reference line in the text.

Comments:
This is a rather elaborate method for addressing errors principally originating due to sensor response. I believe the authors have made a compelling case for the merits of this method but they haven't demonstrated its performance relative to much simpler methods designed to correct for hysteresis (e.g., Miloshevich et al. 2004). This is the 800 lb gorilla in the room and it really should be addressed using both the synthetic and actual data included in their evaluation.

Line 25: It's probably worth emphasizing that the measurement errors due to the "turbulent micro-environment around the body" are much more of an issue with multi-rotor UAS than fixed-wings.

Line 216: Need to cite Waugh (2021) here.

Line 233: If temperature changes significantly across a RH shock (which is often the case), doesn't this add to the error when assuming a mean temperature? Is this dealt with during optimization?

Line 402: An airspeed of 45 m/s for small fixed-wing UAS is not typical. I've worked with a number of FW sUAS and they all operate at airspeeds around 15-30 m/s. Sure, some of them can fly 45 m/s but this operation mode is in the tails of the distribution.

Line 423: Need to include the FAA authorization under which data were collected (e.g., COA number or "Part 107" [including exemptions required to operate up to 1300 m AGL]).

Miloshevich, L. M., Paukkunen, A., Vömel, H., & Oltmans, S. J. (2004). Development and Validation of a Time-Lag Correction for Vaisala Radiosonde Humidity Measurements, Journal of Atmospheric and Oceanic Technology, 21(9), 1305-1327.

Waugh, S. M. (2021). The "U-Tube": An Improved Aspirated Temperature System for Mobile Meteorological Observations, Especially in Severe Weather, Journal of Atmospheric and Oceanic Technology, 38(9), 1477-1489.

---

## Author Comment (AC1)

The authors would like to thank the reviewers and editors for their insightful questions and feedback. These comments have undoubtedly improved the quality of this manuscript. Author responses to each individual comment are outlined below.

Author's response to Anonymous Referee #2
Referee's comments are bold and italicized, and the author's responses are plain text.

***The discussed topic is very important for atmospheric measurement on moving platforms and using -IDMP can improve the data quality significantly. But the paper itself shows lots of shortcomings:***

***1. Does the paper address relevant scientific questions within the scope of AMT?***

***Yes, definitely.***

Thank you for the positive feedback.

***2. Does the paper present novel concepts, ideas, tools, or data?***

***The paper discusses the inverse dynamic modell processing (IDMP) approach to correct dynamic response errors in the measured signal using simulated input data. This concept is important to improve real inflight data and is a current topic of discussion.***

Thanks for your positive comments, not only is the paper a discussion of the IDMP but also a reference to the collection of considerations and conditions to take into account so that the IDMP delivers good quality results. Additionally, this study also provides contributions on model stability analysis necessary for proper parameter tuning of this particular kind of sensor measurement correction method.

***3. Are substantial conclusions reached?***

***An overview about the topic of thermos-hygrometer-sensors is given and the IDMP approach is well discussed. While the IDMP is deeply discussed other topics like installation aspects and static sensor errors are only dealt within a short literature research.***

Thanks for the positive comments. Regarding the concern about the short literature review, we agree with the reviewer that the sensor characterization and installation topics were not directly addressed by the authors with real evidence and experimentation. However, we think that if we leave this topic open ended, then it could potentially lead to a misuse of the presented IDMP method. Therefore, we put together a collection of citations and literature review that guides the reader towards best practices on sensor characterization and placement on a UAS. The authors thoroughly revised the literature and put it in context with

the citation provided. Additionally, many of the literature research cited in chapters 4 and 5 were mostly collaborations that we made with other researchers and institutions. We used those studies to make considerations and produce the best possible conditions for the IDMP application.

*4. Are the scientific methods and assumptions valid and clearly outlined?*

*Chapter 2 is listing some theory about turbulence but without a proper discussion about the implications of each equation for this paper and without a discussion about the applicability.*

The authors agree with this concern. After thoroughly revising the text, we realized about the inconsistency of the narration and lack of implications towards the presented study. Therefore, we edited chapter 2 to reflect the implications of the assumptions and equations to our study as well as its applicability on the IDMP method. The new parts now read:

Added in Line 70: *"Additionally, UASs typically fly in regions of the atmosphere that are hardly accessible to other conventional weather instruments, and collocated intercomparisons are hard to achieve. Therefore, the authors have resorted and adopted other ways of validation, such as the spectrum analysis, which requires particular atmospheric conditions and some theoretical background."*

Added in Line 84: *"Assuming the volume of air sampled by the UAS is locally isotropic, then the thermo-hygrometer sensors should observe a pattern similar to the -5/3 slope line. Assuming that the frequency content of the atmospheric eddies are larger than the frequency range that the sensor can capture, then any deviation in the sensor-data spectrum was assumed to be influenced by undesired sensor dynamics or noise."*

Added in Line 99: *"The computation of the structure function is straightforward and has relatively less theoretical assumptions than the conventional spectral analysis (Gibbs et al., 2016). Therefore, the structure function was considered as an extra step in the validation of the thermo-hygrometer observations and dynamical analysis."*

Overall, this chapter shows a summary of methods to indirectly measure the validity of the thermo-hygrometer sensors observations aboard a UAS in real world conditions, where a direct comparison with another instrument is difficult if not impossible.

*The IDMP is a well suited method and discussed in depth.*

Thank you for the positive feedback.

*The thermodynamic system equations do not include any of the real world installation constrains described in Ch. 5. I shall be discussed if radiation, heat conductivity and heat capacity effects can be neglected.*

The reviewer is correct, we recognize that we failed to clarify this in the paper. The equations described for each sensor only correct for the internal diffusion dynamics of temperature and humidity respectively. We assumed this to be the main contribution of the measurement discrepancies if the considerations of sensor placement and characterization are followed properly. Therefore, other types of contaminations were considered to be negligible. To fix this issue, we modified lines 115-119 to describe our reasoning, it now reads:

*"Moreover, Greene et al. (2019) has shown us that a good shielding around the sensors and an adequate sensor placement on the UAS can greatly prevent solar radiation and heat conduction from contaminating the sampled volume of air. Therefore, for this study, the external sources of contamination are considered negligible compared to the errors introduced by the internal sensor dynamics."*

*"Wildmann et al. (2014b) used a similar approach and showed an example of the modeling of a capacitive humidity sensor using the diffusion equation and effectively applying an inverse model to correct the measurements. We were able to reproduce their modeling methods and validate the results with similar simulation experiments. Given the simplicity of this method, ideas from Wildmann et al. (2014b) studies were borrowed to develop the IDMP proposed in this study and it also served as a guidance to develop an IDMP variant for the bead thermistor. We also realized that a stability analysis with parameter tuning of the models, shown in Section 6.2, would be a great complement to their studies, and a necessary tool for correct application of the IDMP"*

*Chapter 4 and 5 are limited literature reviews without proper discussions about the applicability for the question at hand. Both chapters do not reflect the in depth IDMP discussion.*

This question has been addressed and answered in Question 3 (see above). A summary is shown here:

We believe that chapters 4 and 5 are important considerations towards the correct use of the IDMP. We recognize that these chapters don't really show in-depth studies and proofs. However, the collection of references and citations are enough information to help put together steps, guidelines and limitations on the use of the IDMP.

*Regarding chapter 2, 4 and 5: If no detailed discussion is aspired, please clarify the assumption drawn from the source as assumptions with all its limitations.*

Thanks for bringing up this concern. We address this issue below:

In regards to chapter 2, we agree with the reviewer that there is a lack of clarification about the assumptions drawn from the sources and their implications in our study. This request has been addressed in Question 4 item 1 (see above), where we included additional content that clarifies the assumptions made and explains the implications towards our study.

In regards to chapter 4 and 5, we agree with the reviewer that the authors could not contribute with direct evidence and support of these assumptions. However, we felt compelled to elaborate a "user guide" with a list of best practices for the correct use of the IDMP method based on an extensive compilation of previous studies (shown as citations). We thoroughly surveyed and revised the literature to provide useful references so that the reader could follow steps for the correct use of the presented method. A few limitations of the sensor characterization and installation were also described in our paper based on the literature review, each one of them has in-depth analysis on their respective scientific paper.

***5. Are the results sufficient to support the interpretations and conclusions?***

***The IDMP signal reconstruction based on simulated data seems to be consistent. Interpreting the used real data lacks on reference data, which would require proven faster and more accurate sensors. Looking at the used sensors reference sensors exist. I guess it was not possible to tailor the experimental setup for this question here, so I would not call chapter 7.4 an evaluation but an application to real data.***

The authors agree with the reviewer, we didn't have other types of sensor at the moment. However, we tried to make an evaluation of the IDMP by sending the UAS through the air mass boundary at different speeds (ascend: 3.5m/s and descent: 6m/s) and make a comparison from there. Please recall that the UAS speed across thermal gradients is another factor that can affect the quality and representativeness of the measurement, which was part of the evaluation although the speed range was small. However, we agree that this was not a fully thorough evaluation, therefore we renamed this section *"Case study using real data"*.

***As apparently better looking data are never a proof, this chapter should be check regarding the conclusions. Best would be to add and discuss hypothesis to be falsified or proven using the real data.***

We agree with the reviewer about the need for other evidence and proof to accept the IDMP as a valid tool for correcting real data. This case study is not enough to fully support the use of the IDMP method. However, the main goal in this chapter is to basically show the feasibility of using the IDMP outside the ideal conditions of the simulations. It is shown that the IDMP remained stable throughout both flights and it even produced corrected weather signals that are more consistent with Kolmogorov's power spectrum theory. Therefore, the Conclusions of the paper were edited to reflect the drawbacks and achievements of the case study, it now reads:

"This document presented an overview of the general procedures for the mitigation of undesired sensor dynamics on temperature and humidity measurements collected using a UAS. Important considerations about the effects of the UAS on the sensor measurements were shown, in which it is encourage to find solutions that benefit both flight stability and weather sampling accuracy.

Furthermore, sensor models were developed to investigate the sensor transient response and a collection of best practices for sensor characterization and installation was presented to ensure reduced contamination of the air sample. This allowed for the design and implementation of signal restoration techniques under adequate conditions, such as the IDMP shown in this study. The authors also took a step further and studied the stability of the sensor models and the IDMP method. A system tuning criteria was presented which helped determine the operating envelope of the IDMP and its limitations. This same analysis could be used as leverage for finding improvements within the sensor model design and the sensor selection for the desired application.

After a brief review of the PBL theory, it has been found ways to create a correction criteria based on the power spectrum density calculation. As a result, the sensor measurements were corrected in a way that the resulting power spectrum was more consistent and aligned to Kolmogorov's power spectrum theory under locally isotropic assumptions. The structure function was then used as a mean to corroborate the corrections which gave some degree of validation to the results. The simulation results served as a good evidence for this criteria where the mitigation of undesired contamination, and signal restoration using the IDMP was found to be significant to improve the reliability of the weather UAS deliverables when flying across strong thermodynamic gradients.

Finally, the case study demonstrated the feasibility of using the IDMP outside of the ideal and simulated conditions. The IDMP remained stable throughout both flights while also making small sensor response corrections in the time domain, which is more noticeable in the frequency domain where it follows the 2/3 slope more consistently. Despite these achievements, the case study is not enough material to fully support the use of the IDMP for sensor measurement correction. However, the case study is considered to be a good trend towards producing weather signals with richer frequency content relative to Kolmogorov's theory that can lead to a better understanding of the atmosphere's structure."

**Chapter 6 and 7 would need a more critical discussion, especially as the method described by Wildmann et al. 2014b would have needed a critical review before.**

The method described by Wildmann et al. 2014b was the main foundation for the IDMP studies. We explicitly say that we borrowed ideas from Wildmann in lines 115-118 to further develop the presented IDMP. However, we decided to make a better description of his contribution to our work in Chapter 3 lines 116-119 that now reads:

*"Wildmann et al. (2014b) used a similar approach and showed an example of the modeling of a capacitive humidity sensor using the diffusion equation and effectively applying an inverse model to correct the measurements. We were able to reproduce their modeling methods and validate the results with similar simulation experiments. Given the simplicity of this method, ideas from Wildmann et al. (2014b) studies were borrowed to develop the IDMP proposed in this study and it also served as a guidance to develop an IDMP variant for the bead thermistor. We also realized that a stability analysis with parameter tuning of the models, shown in Section 6.2, would be a great complement to their studies, and a necessary tool for correct application of the IDMP."*

We also referenced his work in other parts of our paper where his contribution was key:

Line 288: *"In order to detail and formulate the IDMP restoration technique, it will be beneficial to start by summarizing a basic general procedure, part of them taken from (Wildmann et al. 2014b):"*

**6. Is the description of experiments and calculations sufficiently complete and precise to allow their reproduction by fellow scientists (traceability of results)?**

**To achieve full traceability the used simulated and real data sets should be published in a data repository and cited here.**

In regards to the simulated data, we believe that our paper thoroughly describes the way to generate the simulated weather signals as shown on Chapter 7.1. Which even allows for flexibility to accommodate other needs and conditions. Therefore, we think that there is no need to share the simulated dataset.

Unfortunately, we don't have the real-world dataset used for this study stored and available in an open repository. However, similar data collected with the same UAS can be found in Elizabeth A Pillar-Little et al. 2021 which was included in Chapter 7.4 lines 436-437 and it reads: "Although these two flights are not available in an open online repository, similar dataset can be found in Greene et al. (2020) and described in Pillar-Little et al. (2021) which were collected with the same UAS in this study"

**7. Do the authors give proper credit to related work and clearly indicate their own new/original contribution?**

**In Chapter 3 it's not clear for me what is cited from Wildmann et al. 2014b and what is the authors contribution. Please clarify.**

This question was addressed in Question 5 item 3 (see above). We decided to make a better description of his contribution to our work in Chapter 3 lines 116-119. We also referenced his work in other parts of our paper where his contribution was key, like in line 288.

**8. Does the title clearly reflect the contents of the paper?**

**The title should name the main topic more clearly, e.g. "Correcting thermos-hygrometer-sensor dynamics …". The term "framework" is no appropriate, as not all aspect are discussed in a proper way.**

After a discussion with the authors, we think that changing the title to *"Considerations for Improving Data Quality of Thermo-Hygrometer Sensors aboard Unmanned Aerial Systems for Planetary Boundary Layer Research"* has a better fit to the presented study. Even though the paper is focused on the applicability of the IDMP, we cannot ignore the collection of considerations and assumptions made and taken from our extensive literature review. Moreover, many of the literature presented in chapters 4 and 5 were actual collaborations that we carried out with other researchers and institutions. We used them as guidelines and as a way to explain the ideal scenarios for the application of the IDMP.

**9. Does the abstract provide a concise and complete summary?**

**The content of the abstract promises a well balance discussion of the title topic. As written above the paper is IDMP focussed and this should be addressed in the abstract as well.**

A few changes in the abstract were made to better reflect and advertise the application of the IDMP, in addition to including a small contribution and a better description of the case study presented. The abstract now reads:

*"Small Unmanned Aerial Systems (UAS) are becoming a good candidate technology for solving the observational gap in the planetary boundary layer (PBL). Additionally, the rapid miniaturization of thermodynamic sensors over the past years allowed for more seamless integration with small UASs and more simple system characterization procedures. However, given that the UAS alters its immediate surrounding air to stay aloft by nature, such integration can introduce several sources of bias and uncertainties to the measurements if not properly accounted for. If weather forecast models were to use UAS measurements, then these errors could significantly impact numerical predictions and, hence, influence the weather forecasters' situational awareness and their ability to issue warnings. Therefore, some considerations for sensor placement are presented in this study as well as flight patterns and strategies to minimize the effects of UAS on the weather sensors. Moreover, advanced modeling techniques and signal processing algorithms are investigated to compensate for slow sensor dynamics. For this study, dynamic models were developed to characterize and assess the transient response of commonly used temperature and humidity sensors. Consequently, an inverse dynamic model processing (IDMP) algorithm that enhances signal restoration is presented and demonstrated on simulated data. This study also provides contributions on model stability analysis necessary for proper parameter tuning of the sensor measurement correction method. A few real case studies are discussed where the application and results of the IDMP through strong thermodynamic*

*gradients of the PBL are shown. The conclusions of this study provide information regarding the effectiveness of the overall process of mitigating undesired distortions in the data sampled with a UAS to help increase the data quality and reliability."*

**10. Is the overall presentation well structured and clear?**

**The overall structure is well understandable.**

Thank you for the positive comment.

**11. Is the language fluent and precise?**

**Yes.**

Thank you for letting us know that our language and narration is transmitting the ideas fluently and precisely.

**12. Are mathematical formulae, symbols, abbreviations, and units correctly defined and used?**

**No mistakes were found.**

Thank you for revising our mathematical formulae.

**13. Should any parts of the paper (text, formulae, figures, tables) be clarified, reduced, combined, or eliminated?**

**Several items to clarify are addressed above.**

**Lines 42 – 66: Please check for a certain redundancy in telling the story.**

After reviewing this section, we agree with the reviewer that the text has redundancies and it can be shortened up with a better narration. Therefore, some modifications were made to this section to remove the redundancies in the text. Line 42-66 now reads:

*"The acquisition of weather data using UAS is a newly established challenge in modern meteorology research, which is slowly showing its potential to create new advanced sampling strategies and signal processing capabilities. The mitigation of slow sensor dynamics and the removal of sensor noise using low cost weather sensors are challenging, but the impacts can be reduced by using the right tools. The inverse dynamic model processing (IDMP) techniques have traditionally been used in control theory for the design of controllers to influence the system's behavior. This modern technique makes use of known physical properties of the sensor to restore the original signal given a sensor reading. To ensure a reliable and proper functioning of the weather sensors, it is important to mitigate sources of error around the UAS by applying strategies discussed in this study,*

*in particular for temperature and humidity sensors. Slow transient response in sensors are commonly associated with amplitude attenuation and phase delay of the output signal (measured weather signal) with respect to the input signal (actual weather signal). While the impact of sensor dynamics can largely be neglected when considering static scenarios, measurements should not be considered instantaneous in space and time when the sensor moves through strong gradients (Houston and Keeler, 2018). Several studies have proposed ways to reduce the impact of the sensor transient response for temperature (Dantzig, 1985; Fatoorehchi et al. 2019) and humidity (Wildmann et al., 2014b) sensors.*

*Considering the above context and problem definition, the following study presents considerations for the sensor characterization and placement on UAS. It also shows a framework for measurement correction of temperature and humidity sensors with data collected using rotary-wing UAS. The goal of this project is to improve the quality of the weather data by following a framework designed around the IDMP method. This will result in a more accurate weather parameter estimates that could, in a near future, improve data assimilation into weather forecast models and, hence, issue accurate weather warnings. It is critical to provide forecasters with reliable data in a timely manner to support them in their mission of protecting lives and properties."*

**14. Are the number and quality of references appropriate?**

*As far as I could check, the chosen references*

Thank you for checking our references. A few additional references were included regarding the real-world dataset availability as requested by the reviewer:

- Greene, B. R., Bell, T. M., Pillar-Little, E. A., Segales, A. R., Britto Hupsel de Azevedo, G., Doyle, W., Tripp, D. D., Kanneganti, S. T., and Chilson, P. B.: University of Oklahoma CopterSonde Files from LAPSE-RATE, https://doi.org/10.5281/zenodo.3737087, 2020.
- Pillar-Little, E. A., Greene, B. R., Lappin, F. M., Bell, T. M., Segales, A. R., de Azevedo, G. B. H., Doyle, W., Kanneganti, S. T., Tripp, D. D., and Chilson, P. B.: Observations of the thermodynamic and kinematic state of the atmospheric boundary layer over the San Luis Valley, CO, using the CopterSonde 2 remotely piloted aircraft system in support of the LAPSE-RATE field campaign, Earth System Science Data,13, 269–280, https://doi.org/10.5194/essd-13-269-2021, 2021.

**15. Is the amount and quality of supplementary material appropriate?**

*n.a.*

**Some further remarks:**

*Line 23: Radiosondes do also affect their sensors, although in a less dramatic amount.*

We agree with the reviewer's statement. We believe that line 23 transmits a similar expression and there's no need for edit.

*Ch2: Please clarify the assumption of a frozen pattern and homogeneous isotropic turbulence.*

This request has been addressed in Question 4 item 1 (see above).

*Line 317f: Please justify this assumption.*

Thanks for bringing up this concern. The reviewer is correct in that the authors must justify the assumptions made for the correction criteria of the IDMP method.

The power spectrum of pure sensor noise is, in general, flat (0 slope) across all frequencies. This is known as the noise floor, inexpensive sensors usually have high noise floor. Subsequently, because Kolmogorov's power spectrum has a constant downward slope of -5/3 across higher frequencies, it will eventually encounter the noise floor of the sensor. Consequently, the weather signal gets buried and lost in the noise. This is manifested as an increase of the power spectrum slope (>-5/3), the inflection point is then used to determine the cutt-off frequency of the low-pass filter.

The opposite to this is due to the poor sensor dynamics that cannot keep up with the turbulence dynamics of the atmosphere. As a result, the signal looks more attenuated (or smoothed out) and sometimes distorted.

Changes were made in this part of the paper, line 317-320 now reads:

*If the slope of the PSD is greater than -5/3 at high wavenumbers, then the region is considered to be contaminated by sensor noise. This is because of the natural downward trend of the turbulence power spectrum that eventually encounters the noise floor of the sensor. Therefore, it must be removed using the lowpass filter before going through the restoration phase to prevent the noise from getting amplified.*

*Otherwise, the measurement is considered to be attenuated and distorted by the the slow sensor dynamics which cannot keep up with the turbulence dynamics. Consequently, the bad trend is corrected in the restoration phase and the new PSD approximates the desired -5/3 slope line.*

---

## Author Comment (AC2)

The authors would like to thank the reviewers and editors for their insightful questions and feedback. These comments have undoubtedly improved the quality of this manuscript. Author responses to each individual comment are outlined below.

Author's response to Anonymous Referee #1
Referee's comments are bold and italicized, and the author's responses are plain text.

***Recommendation:***
***Accept with minor revisions***

***Summary:***
***The authors present a method for correcting temperature/RH observations collected by small UAS with particular focus on addressing errors associated with sensor response. The method is described thoroughly and includes the motivations and justifications for the decisions made. The method is tested using both synthetic and actual data and performance is evaluated using appropriate techniques.***

***Overall, this is a solid manuscript describing a method for addressing a common source of measurement error applicable to many observation platforms. I have one "major" comment listed below with minor comments listed according to a reference line in the text.***

We thank you for the positive summary. We'll address your concerns below.

***Comments:***
***This is a rather elaborate method for addressing errors principally originating due to sensor response. I believe the authors have made a compelling case for the merits of this method but they haven't demonstrated its performance relative to much simpler methods designed to correct for hysteresis (e.g., Miloshevich et al. 2004). This is the 800 lb gorilla in the room and it really should be addressed using both the synthetic and actual data included in their evaluation.***

Thank you for the positive feedback, we appreciate your suggestion about comparing the performance of the IDMP with other methods. The authors were aware that the paper does not show enough evidence to support the performance of the IDMP method in real-world conditions. However, we believe it was enough to show the feasibility of using the IDMP outside the ideal conditions of the simulations. Based on the presented case study, it was shown that the IDMP remained stable throughout both flights, hence, proving to be a candidate solution. From this point forward, we are going to work on improving the method and prepare it for real intercomparison experiments with other methods which can be a perfect sequel for this study to be addressed in another paper.

***Line 25: It's probably worth emphasizing that the measurement errors due to the "turbulent micro-environment around the body" are much more of an issue with multi-rotor UAS than fixed-wings.***

The reviewer is correct, the authors failed to specify the type of UAS that is affected by turbulence on a larger scale. We decided to modify this sentence, which now reads: "Radiosondes have the advantage that their sensors are exposed to the medium they are sampling without much disturbances, as opposed to their Unmanned Aerial Systems (UAS) counterparts which produce an inherent turbulent micro-environment around its body (Greene et al.,2018). This issue is particularly more severe for multi-rotor UAS compared to fixed-wing UAS."

***Line 216: Need to cite Waugh (2021) here.***

We appreciate the literature recommendation. After reading the paper, we think that the citation to this paper will fit better in chapter 5 "Considerations for Sensor Placement on UAS" since the paper has relevant information about radiation shield design as well as a guide for sensor placement and installation.

***Line 233: If temperature changes significantly across a RH shock (which is often the case), doesn't this add to the error when assuming a mean temperature? Is this dealt with during optimization?***

We agree with the reviewer about the errors introduced into the RH readings by the thermal shock. We think that the explanation given in lines 228-236 is enough to answer the first question. In regards to the second question, the proposed solution was also given in the above-mentioned lines. In other words, the water vapor diffusivity is a function of RH and temperature. Therefore, for every RH shock step, the experiment must be repeated at different air temperatures to create a lookup table (or matrix). Interpolation can be used in between these values.

***Line 402: An airspeed of 45 m/s for small fixed-wing UAS is not typical. I've worked with a number of FW sUAS and they all operate at airspeeds around 15-30 m/s. Sure, some of them can fly 45 m/s but this operation mode is in the tails of the distribution.***

We think the reviewer is correct in that it is hard for a sUAS to reach speeds over 30m/s. However, we think that a good evaluation of the method also includes testing it close or even outside the operating envelope of the sUAS. Seeing satisfactory results under these extreme speeds means that the method can easily handle nominal operating conditions.

***Line 423: Need to include the FAA authorization under which data were collected (e.g., COA number or "Part 107" [including exemptions required to operate up to 1300 m AGL]).***

Thank you for letting us know about this missing information in the paper. Given the high altitude in which the sUAS was flying, it is important to state that all the flights were carried out using a valid Part 107 license with an approved COA from the FAA authorities. The COA information was included in the revised paper, the lines 421-424 now read:

*"These flights were conducted in CBL and FTI weather conditions, respectively, at the Kessler Atmospheric and Ecological Field Station (KAEFS) in Purcell, Oklahoma, USA, located 30km southwest of the OU Norman campus. The Certificate of Authorization (COA) with number 2020-CSA-6030-COA, issued by the Federal Aviation Administration (FAA), allowed us to fly the CopterSonde above 400ft with a flight ceiling of 5000ft."*

**Miloshevich, L. M., Paukkunen, A., Vömel, H., & Oltmans, S. J. (2004). Development and Validation of a Time-Lag Correction for Vaisala Radiosonde Humidity Measurements, Journal of Atmospheric and Oceanic Technology, 21(9), 1305-1327.**

Thank you for the reference.

**Waugh, S. M. (2021). The "U-Tube": An Improved Aspirated Temperature System for Mobile Meteorological Observations, Especially in Severe Weather, Journal of Atmospheric and Oceanic Technology, 38(9), 1477-1489.**

Thank you for the reference.